# Propagation and Diffusion of Fluorescent Substances with Footprints in Indoor Environments

**DOI:** 10.3390/ijerph19137733

**Published:** 2022-06-24

**Authors:** Manman Ma, Fei Li, Hao Han, Ziwang Zhao, Yuxiao Sun, Yuanqi Jing, Lei Wang

**Affiliations:** 1State Key Laboratory of NBC Protection for Civilian, Beijing 100191, China; mamanman0608@163.com (M.M.); zhaoziwang1123@163.com (Z.Z.); alvin_leilei@126.com (L.W.); 2College of Urban Construction, Nanjing Tech University, Nanjing 211816, China; syx17826265985@163.com (Y.S.); jingyuanqi@njtech.edu.cn (Y.J.)

**Keywords:** fluorescent substances, pathogens, ground pollutant diffusion, footprint

## Abstract

Some studies have shown that contaminants can be transferred between floors and the soles, and there are few studies on pollutant propagation caused by human walking in real-life situations. This study explored the propagation and diffusion law of ground pollutants from rubber soles to poly vinyl chloride (PVC) floor during indoor walking through employing a fluorescent solution as a simulant. The footprint decay (*D*) and transfer efficiency (*τ*) of the fluorescent solution transferred from the sole to the indoor floor during walking were analyzed based on the fluorescent footprint imaging. The effects of namely body weight (50–75 kg), walking frequency (80–120 steps/min), and solution viscosity (oil and water) were also investigated. It was found that the total fluorescence gray value on the ground decreased exponentially as the number of walking steps (*i*) increased. The relationship between the normalized gray value of the fluorescent solution (*D*) on each floor panel *i* was $D_{i}=ae^{bi},\left(2.1\leq a\leq 3.8,–1.4\leq b\leq –0.7\right)$, and *τ* was distributed in the range of 0.51–0.72. All influencing factors had a significant effect on *a*, and a greater body weight resulted in a smaller *a* value, while only the body weight had a significant effect on *b* and *τ*, and a greater body weight led to larger *b* and lower *τ* values.

## 1. Introduction

In the past few years, the novel coronavirus (COVID-19) outbreak has propagated worldwide, causing massive impacts on the economic and social stabilities of the affected countries [1,2,3]. Currently, the recognized pathogen propagation modes are through droplets, aerosols, physical contact, and the fecal−oral route [4,5,6], with the majority of research focusing on droplet and aerosol propagation. However, when samples were taken from surfaces in a hospital environment accommodating coronavirus patients, positive test samples were recorded from floors, computer mice, trash cans, and patient bedrails [7]. Surprisingly, a positive detection rate of 100% was also recorded on the pharmacy floor, to which the patients had no access [8]. Thus, in view of the fact that the pharmacy floor is contacted by medical staff alone, and studies have shown that contaminants can be transferred between floors and the soles of shoes [9], there is a possibility that the detected pathogens were brought into the pharmacy on the shoes of medical staff, ultimately propagating their spread. However, to date, this pathogen propagation mode has not received particular attention.

Air quality issues have received widespread attention in recent years [10,11], and there have been several studies on the exposure transmission of pollutants. Therefore, understanding the factors that affect the transfer efficiency during contact would be expected to have important implications for containing propagation. To analyze the relationship between the microbial transfer rate and influencing factors during hand touching, Zhao et al. [12] used the principal component analysis and found that the donor humidity, microbial transfer direction, donor roughness, and friction were the most significant factors. In addition, Xiao et al. [13] used fluorescence imaging technology to quantify fluorescent particles on the surfaces and studied the dynamic transport process of the norovirus on aircraft through hand contact on multiple environmental surfaces. Currently, the majority of contact propagation studies focus on the transfer laws from hand to surface or from surface to surface [14,15,16,17,18]. Sippola et al. [9] measured the down lay transfer efficiency of particles between shoes and floors to understand the transport of deposited particulate pollutants, and Rosati et al. [19] measured the single-step absorption transfer efficiency of amorphous silica particles from carpets to shoes. Similarly, Hunt et al. [20] measured the transfer of wet and dry soil from shoe soles to tiles and found that the transfer efficiency decreased with each successive step. Furthermore, Mcdonagh et al. [21] quantified the mass transfer efficiency of aerosol particles deposited on hard and soft surfaces by changing the applied pressure, contact time, and pollutant loading, and found that the transfer efficiency did not increase significantly with an increase in contact time. As the applied pressure and surface roughness increased, the transfer efficiency also increased. Although some studies have investigated pollutant propagation between shoe soles and the ground, the studies on pollutant propagation and diffusion caused by human walking in real-life situations are lacking. Moreover, the pathogens can also exist in the saliva and mucous sputum of personnel, and few studies have been carried out on the spread of liquid pollutants on shoe soles.

In this study, the propagation and diffusion law of ground pollutants with footprints during indoor walking were investigated through employing a fluorescent solution and footprint imaging. More specifically, these tests were carried out based on the fluorescent solution from shoe soles onto the indoor ground via walking. The influence of three factors, namely body weight, walking frequency, and solution viscosity on the footprint decay and transfer efficiency were examined in detail. Finally, the diffusion law was determined and the factors influencing this type of diffusion mode were clarified.

## 2. Experimental and Analytical Methods

Experiments were based on a charge-coupled device and used violet light-excited fluorescence to obtain fluorescent footprint images to measure the amount of fluorescent solution transferred to the floor panel via walking. Each experiment had three phases: (1) fluorescent solution preparation, (2) exposure process, and (3) fluorescence data analysis.

### 2.1. Selection and Preparation of the Fluorescent Solution

Green phosphor (MJ-818, 3–5 μm particle size), which consisted of a non-toxic amino resin pigment (CH_3_C_6_H_4_SO_2_NH_2_[CH_2_O]_n_[NH_2_]_3_N_3_), was purchased from Guangzhou Qihong Technology Co., Ltd., Guangzhou, China. The emission and excitation spectra of the industrial phosphor were concentrated in the range of 350–650 nm.

In this study, edible soybean oil was used as a simulation liquid to replace sputum because its viscosity at room temperature is 57.1 mPa·s, which is similar to that of the sputum of patients with lung disease (52 mPa·s) [22]. Moreover, as the main component of saliva is water (99–99.5%) [23], water was used to replace saliva (water viscosity at room temperature = 1 mPa·s).

The fluorescent solution was prepared according to the following procedure. First, different masses of an industrial water-based fluorescent powder (3 and 8 g) were added to the water and edible soybean oil (100 mL each), respectively. Then, each mixture was magnetically stirred for 10 min until the solid phosphor particles were completely dissolved or dispersed. Subsequently, an aqueous-based phosphor solution was prepared with a concentration of 0.03 g/mL, while an oil-based solution was prepared with a concentration of 0.08 g/mL. Equal amounts of the two solutions were poured into an experimental tray to ensure they were evenly dispersed at the bottom of the tray and that the height of the solution was sufficient to completely cover the bottom of the tray. Then, the experimental tray was placed on a horizontal surface for later use.

### 2.2. Selection of the Experimental Floor and Shoes

The experimental floor panels were composed of polyvinyl chloride (PVC), which is commonly used in hospitals. The floor panels were classed as “water blue” in color, and the panel surfaces were coated with an ultraviolet antifouling layer that was easy to clean. Each experimental floor panel was cut into a rectangle measuring 25 × 40 cm to ensure that it could accommodate the single foot of an average adult male. The experimental floor panels were vertically staggered in a corridor with a spacing of 55 cm, with 11 on each side (left and right; 22 in total), as shown in Figure 1.

The experimental shoes for the female participants were nursing shoes commonly used in hospitals. The sole material was soft rubber with dot-like protrusions, the shoe size was 230 mm, the contact area of the sole was 20 cm^2^, and the tread depth was 2 mm. The experimental shoes for the male participants were casual canvas shoes with rubber soles. The soles exhibited obliquely staggered strip patterns, the shoe size was 260 mm, the contact area of the sole was 35 cm^2^, and the tread depth was 2 mm. Photographic images of the soles of both shoe samples can be seen in Appendix A.

Prior to each experiment, the experimental floor panels and shoes were washed in advance using water and were air dried for 24 h.

### 2.3. Fluorescence Data Collection

#### 2.3.1. Fluorescence Image Acquisition Device

An 80 × 80 × 80 cm camera obscura was built to photograph the contaminated floor panels. As shown in Figure 2, two 60 cm long 20 W violet light tubes were placed on both sides of the top of the camera obscura, 75 cm from the experimental floor panels. The tube lamps emitted ultraviolet light with a wavelength of 365 nm to excite the phosphors on the dyed floor panels. A charge-coupled device camera named Insta360 (Arashi Vision, Shenzhen, China) was fixed in the center of the camera obscura, the fluorescence images were photographed, and the camera was set to fix the photographing parameters.

To explore whether the light angle and intensity had any effect on the fluorescence intensity of the fluorescent solution on the experimental floor panels, a paired sample *t*-test was used to assess the influence of different positions in the dark box and the number of violet lamps that were switched on. Changing the light angle in the dark box had no statistical effect on the average gray value of the fluorescent solution on the floor panels (*p* = 0.167). However, the effect of changing the light intensity on the average gray value of the fluorescent solution on the floor panels showed a significant difference at the 0.05 significance level. Therefore, during fluorescence data collection, the same number of violet lights must be switched on in each case, and the frequency should be stable (i.e., ~5 min after switching on).

#### 2.3.2. Image Processing and Data Extraction

After walking, the contaminated floor panels were placed into the camera obscura in order of stepping, and the camera obscura was fully closed during recording of the images. ImageJ software was used to uniformly crop the captured images to 1200 × 700 pixels and then perform gray processing. To exclude the interference of the background fluorescence in the experiment, the threshold was set to a background-specific value of 70–255 (see Figure 3). Using this method, the total grey value of the fluorescent solution on each experimental floor panel was obtained, which is correlated with the total amount of fluorescent substances. The calculation method for obtaining the total and average gray values is as follows:(1)$$\begin{array}{c} Intden=\sum_{i=70}^{255}i\left(x,y\right),\;\left(0\leq x\leq 1200,0\leq y\leq 700\right)\\ MeanIntden=\frac{Intden}{Area} \end{array}$$
where *Intden* represents the total gray value, *i*(*x*, *y*) represents the gray value of the (*x*, *y*) pixel point, *MeanIntden* represents the average gray value, and *Area* represents the pixel area of the selected area.

#### 2.3.3. Statistical Analysis

The Origin2017 (OriginLab Corp., Northampton, MA, USA) and SPSS 23 (IBM Corp., Armonk, NY, USA) software packages were used for plotting and data analysis, respectively. The paired sample *t*-test was employed to determine whether there was a statistically significant difference in the amount of fluorescent solution on the floor panels between the data collection conditions and the left- and right-foot walking characteristics of the participants. Analysis of variance (ANOVA) was used to determine whether the body weight, walking frequency, and solution viscosity resulted in statistically significant differences in the transfer efficiency and the amount of fluorescent solution on the floor panels.

### 2.4. Experimental Case Design

According to the Report on Nutrition and Chronic Diseases in China (2020) [24], the average weight of residents aged 18 years and above was 69.6 kg for males and 59 kg for females. For the experiment, three volunteers were recruited, and their weight ranges were 50–55 kg, 60–65 kg, and 70–75 kg, taking into account the regional differences and weight fluctuations. Volunteer No. 1 was a woman weighing 54 kg with a shoe size of 230 mm, Volunteer No. 2 was a man weighing 63 kg with a shoe size of 260 mm, and Volunteer No. 3 was a man weighing 72 kg with a shoe size of 260 mm. The walking frequency was controlled using a metronome and was set to 80, 100, and 120 steps/min to represent slow, normal, and fast walking states, respectively.

At the beginning of the experiment, the participants wore the experimental shoes and stepped up and down alternately in the experimental tray containing the fluorescent solution to ensure that the shoe soles were completely covered by the fluorescent solution. Subsequently, under the set walking frequency, the participants took the first step with their left foot on the first experimental floor panel prior to continuing along the panel route presented in Figure 1.

The experimental design adopted three factors and three levels in an orthogonal layout. Table 1 outlines the three influencing factors: body weight (three levels), walking frequency (three levels), and solution viscosity (two levels). As the solution viscosity only had two levels, the dummy level method was used for processing. Each experiment was repeated six times.

## 3. Results and Discussion

### 3.1. Single-Footprint Feature Analysis

A contaminated floor panel with a single footprint is shown in Figure 4a, while Figure 4b shows a plot of the change in the average fluorescence gray value across the horizontal pixel points shown in Figure 4a. It can be observed from Figure 4b that the gray plot possesses two peaks and one valley, wherein the peaks correspond with the center of the forefoot and the back of the sole, which are the main contact points during walking. Thus, as expected, the amount of transferred fluorescent solution was at its highest in these areas. Correspondingly, the valley location corresponded to the arch of the foot, wherein the amount of transferred fluorescent substance was lower.

The single-footprint stepping experimental fluorescence images are illustrated in Appendix A. As shown, upon increasing the number of walking steps, the amount of fluorescent substance on the experimental floor panels gradually decreased. The first step resulted in the greatest amount of transfer, wherein the fluorescent substance was mainly concentrated around the contact point between the sole and ground. In addition, numerous connections of the fluorescent solution can be seen in the center of the forefoot and rear heel, where the compression deformation was the largest. As the number of steps increased, the number of connections decreased rapidly. After the fifth step, the amount of overall transferred fluorescence was already low, and only localized punctate fluorescence was visible, which was increasingly difficult to distinguish from the floor panel background.

### 3.2. Analysis of the Independence of Each Foot

To explore how the amount of fluorescent solution on the floor panels was affected by walking on different feet (i.e., left and right feet), three volunteers with a walking frequency of 100 steps/min were selected for the experiment. Based on the results, statistical analysis was carried out, and a plot was obtained to indicate the change in the average gray value between the left and right foot over 11 steps, as shown in Figure 5.

A paired sample *t*-test was used to assess the effect of the difference in walking between the left and right feet, wherein the average gray value of the fluorescent solution was determined for each step number and each volunteer, as can be seen from Figure 5 and Table 2. For each volunteer, it was found that the difference between the left and right feet had no statistically significant effect on the amount of fluorescent solution transferred to the floor panel (*p* > 0.05); therefore, the walking difference between the left foot and right foot was not considered as an influencing factor during the data analysis.

### 3.3. Analysis of the Variation Law of the Ground Fluorescence Value with Walking

The normalized gray value of the fluorescent solution is defined as:(2)$$D_{i}=\frac{G_{i}}{G_{1}},$$
where $G_{i}\;$ represents the total gray value of the fluorescent solution on the *i*-th floor panel, $G_{1}$ represents the total gray value of the fluorescent solution on the first floor panel, and the subscript *i* represents the number of walking steps.

The fluorescence solution on the surface of each experimental floor panel was quantified, as shown in Figure 6a and Appendix A. As indicated, in each group of experiments, an increasing number of steps resulted in a change in the total fluorescence value of the fluorescent solution on the experimental floor panels, wherein a downward trend was observed in each case. Over the first five steps, the total fluorescence value of the floor panel decreased significantly with the number of steps. When the walking frequency and solution viscosity were constant, a lower body weight resulted in a greater attenuation. For the orthogonal experiment carried out using body weights in the range of 50–55 and 60–65 kg, it was found that when the body weight and solution viscosity remained unchanged, a greater walking frequency led to a lower attenuation of the total floor panel fluorescence value. As shown in Figure 6a, the fluorescence value decreased to ~6% of the initial value and began to fluctuate after the fifth step. This was because the quality of the transferred contaminant decreased to a certain low level, and it would be difficult to distinguish the fluorescent substance from the background. In addition, if we knew the human step data, the footprint transmission distance of contaminants could be deduced.

Subsequently, the orthogonal experimental data were exponentially fitted, and the results are presented in Figure 6b and Table 3. For similar work, Zhao et al. [25] investigated the percentage of solid particles remaining on the steel surface after contacting the silicon surface using a pressure-controlled touch machine. They also found a decreasing trend as the number of contacts increased. However, the transfer process in Zhao’s study was mainly in the first three steps, while in our study, the process was mainly in the first five steps. We assumed this was because Zhao’s experiment was conducted using solid particles as the simulant, while our experiment was conducted using a fluorescent solution. Their propagation laws were different. The relationship between the normalized gray value of the fluorescent solution (*D*) on each floor panel *i* was $D_{i}=G_{i}/G_{1}=ae^{bi},\left(2.1\leq a\leq 3.8,–1.4\leq b\leq –0.7\right)$. From the fitted exponential model, it is clear that all R^2^ values were >99%, indicating that the exponential decay model can suitably characterize the propagation and diffusion laws of this system; therefore, it can be used to estimate the number of pollutants on the ground. 

### 3.4. Analysis of the Propagation and Diffusion Influencing Factors

A non-interaction ANOVA was used to analyze the effects of body weight, walking frequency, and solution viscosity on parameters *a* and *b* of the fitting results. As outlined in Table 4, our results show that for parameter *a*, all three influencing factors showed differences at the 0.05 significance level (*p* < 0.05). Among them, body weight was found to be the most important factor, solution viscosity was the secondary factor, and walking frequency had the least impact. When the walking frequency and solution viscosity remained unchanged, a greater body weight resulted in a smaller *a* value; in the case of a constant body weight and solution viscosity, a higher walking frequency led to a smaller *a* value.

Furthermore, for parameter *b* (Table 5), only body weight showed a difference at the 0.05 significance level (*p* = 0.037), wherein the value of *b* increased with body weight. Although no statistical difference was detected for the influence of walking frequency on *b*, we found that *b* decreased with an increasing walking frequency at a constant body weight.

### 3.5. Analysis of the Transfer Efficiency and Impact

#### 3.5.1. Contact Propagation Transfer Efficiency

For the purpose of this study, it was considered that the fluorescent solution was evenly mixed, i.e., the fluorescent solution on all parts of the contactable surface of the sole had the same probability of being transferred to the experimental floor panel. During this contact process, the physical parameters remain unchanged, and thus it was assumed that the transfer efficiency, *τ*, of the fluorescent solution was constant during the continuous stepping process, and that the fluorescent solution on the effective contact area of the sole was completely transferred. Zhao et al. [26] confirmed this hypothesis in the process of sequential hand contact, and *τ* was calculated as follows:(3)$$\begin{array}{c} \tau =\frac{G_{1}}{G_{s1}}=\frac{G_{i}}{G_{si}}=\ldots =\frac{G_{N}}{G_{sN}}\\ G_{n}=\left[\tau {\left(1-\tau \right)}^{n-1}\right]\cdot \left[\frac{\sum_{i=1}^{N}G_{i}}{1-{\left(1-\tau \right)}^{N}}\right]=\frac{\tau {\left(1-\tau \right)}^{n-1}}{1-{\left(1-\tau \right)}^{N}}\left(G_{1}+G_{2}+\cdots +G_{N}\right), \end{array}$$
where *τ* represents the transfer efficiency of the fluorescent substance during the single contact between the sole and the experimental floor panel; $G_{i}$ is the total fluorescence gray value on the *i*-th experimental floor panel after stepping on it, where the subscript *i* is the number of walking steps; $G_{si}$ is the total fluorescence gray value of the sole before each step on the experimental floor panel; $G_{s1}$ is the total fluorescence gray value initially carried by the sole; and $G_{n}$ represents the total fluorescence gray value on the *n*-th experimental floor panel, where 1≤n≤N.

Once the total fluorescence gray value of each experimental floor panel was known, the least-squares method could be used to obtain the transfer efficiency of each contact from the sole to the experimental floor panel according to Equation (3). The results of these experiments and calculations are presented in Figure 7, wherein it can be seen that *τ* is between 0.51 and 0.72, and the correlation coefficient R^2^ is >99%; this model is therefore suitable for characterizing the transfer process of liquid pollutants from the shoe sole to the ground. For similar work, Zhao et al. [25] found transfer rates that were in the range of 0.09–0.21 for *S. aureus* from fingers to glass pieces. McDonagh et al. [21] found it was in the range of 0.02–0.45 for aerosol particles on soft and hard surfaces, while our study found transfer rates were in the range of 0.51–0.72 for the fluorescent solution from rubber soles to PVC flooring. We believe that the reason for the difference is mainly due to the difference in contact surfaces and simulants.

#### 3.5.2. Analysis of the Factors Influencing the Transfer Efficiency

Considering *τ* as the index, the influences of body weight, walking frequency, and solution viscosity were analyzed via variance analysis. As can be seen from Table 6, only body weight had a significant effect on *τ* at the 0.05 significance level (*p* = 0.011), and neither the walking frequency nor the solution viscosity had a significant effect. When the walking frequency and solution viscosity were maintained at a constant, a greater body weight resulted in a lower transfer efficiency. Furthermore, when the body weight and solution viscosity were constant, a higher walking frequency resulted in a lower transfer efficiency (with the exception of the 60–120–W experiment).

## 4. Conclusions

This study used a fluorescent solution to simulate the pollutant propagation from rubber soles to PVC floor during indoor walking and explored the diffusion law of ground pollutants with footprints. The effects of body weight (50–75 kg), walking frequency (80–120 steps/min), and solution viscosity (oil and water) on the footprint decay (*D*) and transfer efficiency (*τ*) of the fluorescent solution were studied via the fluorescent footprint imaging based on a charge-coupled device and using violet light-excited fluorescence. The conclusions are follows:(1)As the number of walking steps increased, the amount of fluorescent substance transferred onto the experimental floor decreased exponentially, the fluorescence value began to fluctuate and decreased to ~6% of the initial value after the fifth step.(2)The relationship between the normalized gray value of the fluorescent solution (*D*) on each floor panel *i* was $D_{i}=G_{i}/G_{1}=ae^{bi},\left(2.1\leq a\leq 3.8,–1.4\leq b\leq –0.7\right)$, and can be used to estimate the number of pollutants on the ground. Based on the variance analysis, it was found that the body weight, walking frequency, and solution viscosity had a significant effect on *a*, with body weight being the most important factor. More specifically, a greater body weight yielded a smaller *a* value. For parameter *b*, only body weight had a significant effect, wherein *b* increased with an increase in body weight.(3)The transfer efficiency was distributed within a range of 0.51–0.72 for all of the experiments, and only the body weight caused it to change significantly. More specifically, with a constant walking frequency and solution viscosity, a greater body weight led to a lower transfer efficiency.

Based on these findings, this work provided a new method for examining the contact propagation and diffusion of ground pollutants while walking. However, the type of real pathogen, its survival time, etc., may have effects on the transmission efficiency. Because of the limitation of the experimental conditions, it is difficult to use the pathogens to conduct the experiment and analyze effects of their number and type. We expect to provide a good guide for subsequent research on the prevention and control of pathogen spread through footprints in a real environment. Additionally, this study is limited by the relatively low detection limit of optical analytical techniques, which cannot distinguish the fluorescent substances from the background floor fluorescence when the quantity of the transferred pollutants is low. Accordingly, further studies should consider using harmless pathogens and higher-sensitivity detection methods.

## Figures and Tables

**Figure 1 ijerph-19-07733-f001:**
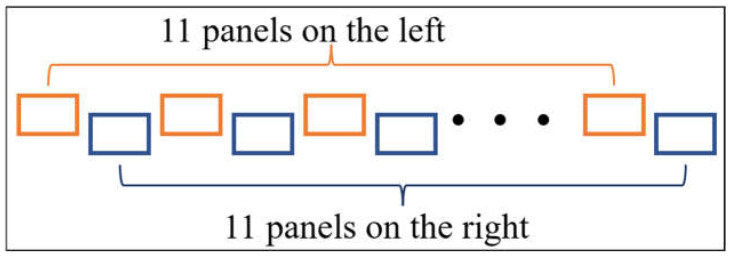
Layout of the experimental floor panels.

**Figure 2 ijerph-19-07733-f002:**
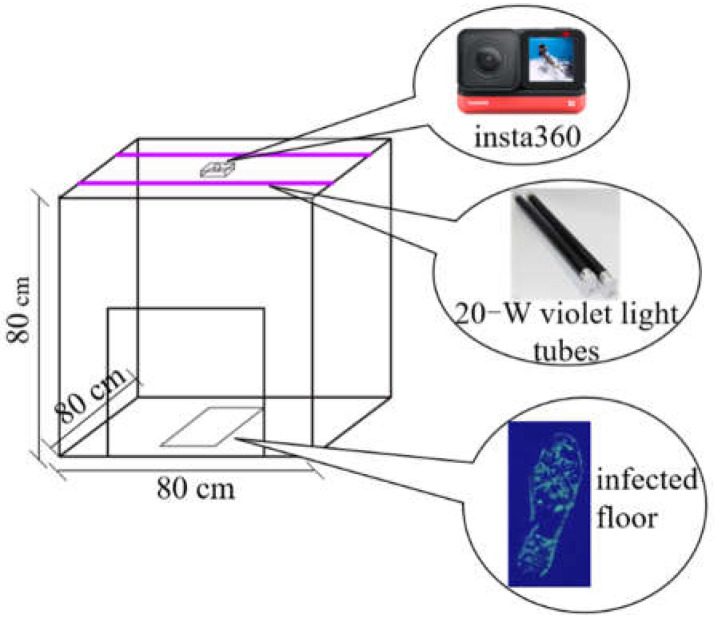
Schematic of the fluorescence image acquisition device.

**Figure 3 ijerph-19-07733-f003:**
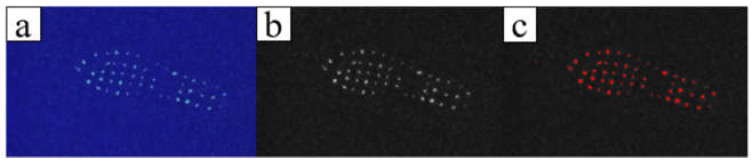
Image processing: (**a**) original fluorescence image, (**b**) gray fluorescence image, and (**c**) threshold range for the selected region.

**Figure 4 ijerph-19-07733-f004:**
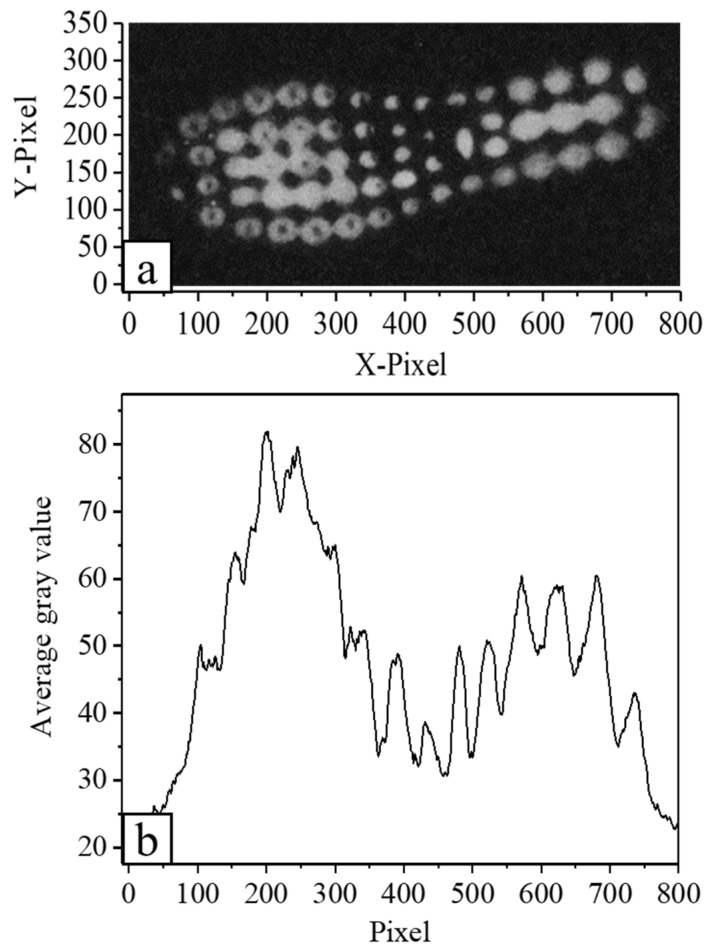
Gray map of single footprint features. (**a**) Gray-scale fluorescence image of a contaminated floor panel. (**b**) Plot showing the change in the average fluorescence gray value across the horizontal pixel points corresponding to (**a**).

**Figure 5 ijerph-19-07733-f005:**
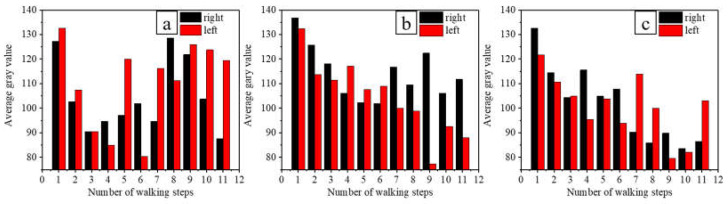
Difference in walking between the left and right feet, wherein the average gray value of the fluorescent solution was determined for each step number. (**a**–**c**) Volunteers No. 1–3, respectively.

**Figure 6 ijerph-19-07733-f006:**
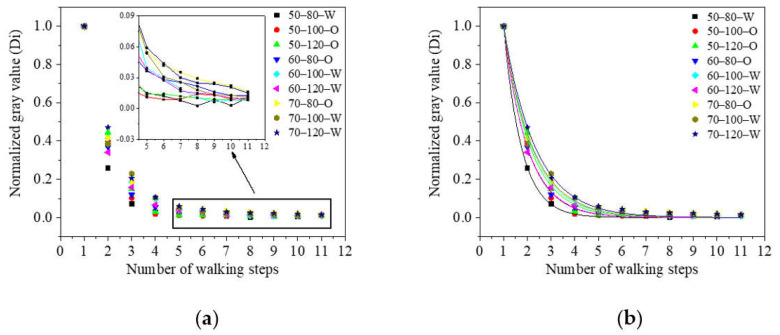
(**a**) Normalized gray value of the fluorescent solution with the number of walking steps. (**b**) Exponential fitting plot of the normalized gray values (50, 60, and 70 represent the weight of 50–55 kg, 60–65 kg, and 70–75 kg, respectively; 80, 100, and 120 represent the walking frequency; W: aqueous solution, O: soybean oil solution.).

**Figure 7 ijerph-19-07733-f007:**
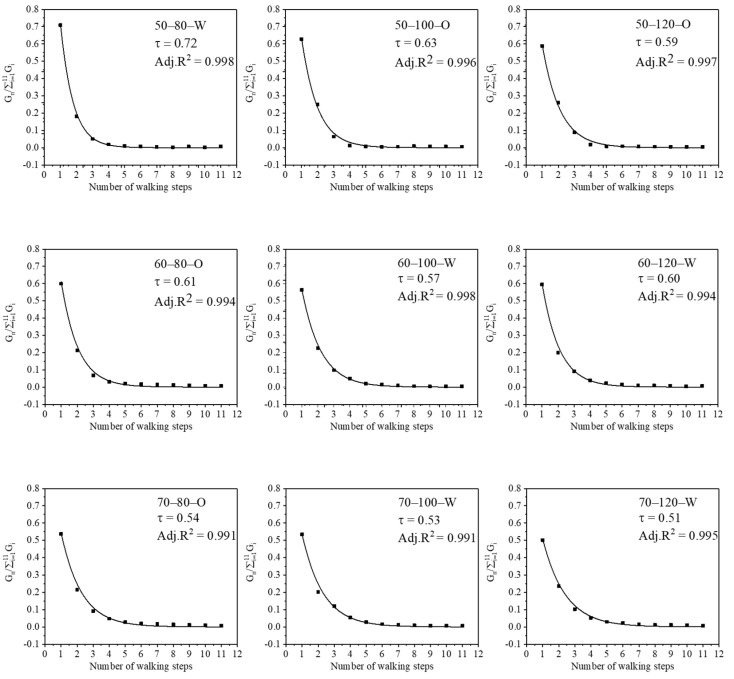
Fitting results for the transfer efficiency *τ*.

**Table 1 ijerph-19-07733-t001:** Orthogonal experimental layout.

Experiment No.	Body Weight (kg)	Walking Frequency (Steps/min)	Solution Viscosity
1	50–55	80	W
2	50–55	100	O
3	50–55	120	O
4	60–65	80	O
5	60–65	100	W
6	60–65	120	W
7	70–75	80	O
8	70–75	100	W
9	70–75	120	W

W = aqueous solution; O = soybean oil solution.

**Table 2 ijerph-19-07733-t002:** Paired sample *t*-test for walking differences between the left and right feet.

Group	*t*	Degrees of Freedom	Significance (Two-Tailed)
Volunteer No. 1	0.668	10	0.519
Volunteer No. 2	−1.234	10	0.246
Volunteer No. 3	1.074	10	0.308

**Table 3 ijerph-19-07733-t003:** Exponential fitting parameters.

Experiment Code	*a*	*b*	R^2^
50–80-W	3.79	−1.33	0.999
50–100-O	2.78	−1.01	0.996
50–120-O	2.48	−0.90	0.996
60–80-O	2.72	−1.00	0.997
60–100-W	2.31	−0.84	0.999
60–120-W	2.68	−0.99	0.996
70–80-O	2.26	−0.82	0.994
70–100-W	2.22	−0.81	0.993
70–120-W	2.11	−0.75	0.997

**Table 4 ijerph-19-07733-t004:** Three-factor ANOVA with *a* as the indicator.

Factor	Sum of Square	*F*	Significant *p*
Body weight	1.340	16.584	0.024 **
Walking frequency	0.785	9.718	0.049 **
Solution viscosity	0.448	11.082	0.045 **

Note: ** indicates a difference at the 0.05 significance level.

**Table 5 ijerph-19-07733-t005:** Three-factor ANOVA with *b* as the indicator.

Factor	Sum of Square	*F*	Significant *p*
Body weight	0.158	11.962	0.037 **
Walking frequency	0.087	6.572	0.080
Solution viscosity	0.044	6.609	0.082

Note: ** indicates a difference at the 0.05 significance level.

**Table 6 ijerph-19-07733-t006:** Three-factor ANOVA using *τ* as the index.

Factor	Sum of Square	*F*	Significant *p*
Body weight	0.025	28.538	0.011 **
Walking frequency	0.008	9.118	0.053
Solution viscosity	0.003	7.803	0.068

Note: ** indicates a difference at the 0.05 significance level.

## Data Availability

The data that support the findings of this study are available from the corresponding author upon reasonable request.

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
