# Peer review of "Propagation and Diffusion of Fluorescent Substances with Footprints in Indoor Environments"

_ijerph, 2022, doi:10.3390/ijerph19137733_

Round 1

Reviewer 1 Report

The authors used a fluorescent solution to simulate the pathogen propagation during indoor walking. In general, the manuscript was well written with both experimental and analytical analyses. The paper could be published after appropriate revisions according to all reviewers’ comments. In the following lines, some suggestions are given for improving the manuscript quality.

  1. A major concern is that the simulated substances on the soles of the feet used in the experiments have high concentrations and uniform distribution, but the actual environmental viral load is typically quite low. How to account for this difference?
  2. Line 18: Please correct the writing error of the total fluorescence gray value on the ground.
  3. Why was the transfer efficiency unchanged during the transfer process?
  4. Line 241: why did fluorescence values fluctuate after the fifth step?
  5. Why were the three weight ranges of 50-55 kg, 60-65 kg and 70-75 kg selected in the experiment?
  6. Why did the authors chose to use charge-coupled device and violet light-excited fluorescence to obtain fluorescent footprint images?
  7. Please further compare your findings with previous studies, such as the experiential fitting curves.

Author Response

(1)、A major concern is that the simulated substances on the soles of the feet used in the experiments have high concentrations and uniform distribution, but the actual environmental viral load is typically quite low. How to account for this difference?

Reply:

As you said, the actual environmental viral load is typically quite low. However, it is difficult to use the pathogens to conduct the experiment due to the experimental conditions. In order to investigate the law of pollutant transmission with footprints in the ground, the fluorescent substance was selected in this study. Although the spread of virus with footprint may have some differences with this experiment, the results of this study can be a good guide for subsequent research on the prevention and control of pathogen spread with footprint in real environment. This discussion has been added to the last paragraph of Section “4. Conclusions”.

(2)、Line 18: Please correct the writing error of the total fluorescence gray value on the ground.

Reply

Thank you very much for your careful review, and the errors have been corrected in Section “Abstract” and “4.Conclusions”: “”.

(3)、Why was the transfer efficiency unchanged during the transfer process?

Reply:

We assumed that if the physical parameters were kept constant during the contact process, the fluorescent substance has the same probability of being transferred from sole to the ground. And this assumption has been verified by a previous study [1]. This explanation has been added to the first paragraph of Section “3.5.1. Contact propagation transfer efficiency”.

References:

[1] Zhao, P.; Li, Y. New sequential‐touch method to determine bacterial contact transfer rate from finger to surface. J APPL MICROBIOL 2019, 127, 605-615.

(4)、Line 241: why did fluorescence values fluctuate after the fifth step?

Reply

When the number of stepping steps increases, the quality of the transferred contaminant will decrease to a certain low level, and it will be difficult to distinguish the fluorescent substance from the background. This will cause fluctuations in the measured values. This discussion has been added to the second paragraph of the Section “3.3. Analysis of the variation law of the ground fluorescence value with walking”.

(5)、Why were the three weight ranges of 50-55 kg, 60-65 kg and 70-75 kg selected in the experiment?

Reply

According to the Report on Nutrition and Chronic Diseases in China (2020), the average weight of residents aged 18 years and above was 69.6 kg for males and 59 kg for females. Taking into account the regional differences and weight fluctuations, the representative weight intervals were set to 50-55 kg, 60-65 kg and 70-75 kg. This discussion has been added to the first paragraph of Section “2.4. Experimental case design”.

(6)、Why did the authors chose to use charge-coupled device and violet light-excited fluorescence to obtain fluorescent footprint images?

Reply:

It is difficult to use the pathogens to conduct the experiment due to the experimental conditions. In order to investigate the law of pollutant transmission with footprints in the ground, the fluorescent substance was selected in this study. Therefore, in this study, we chose charge-coupled device and using violet light-excited fluorescence to obtain fluorescent footprint images. This analysis method can visually obtain the distribution and amount of contaminant on the contact surface directly and rapidly.

(7)、Please further compare your findings with previous studies, such as the experiential fitting curves.

Reply

Zhao et al. [1] investigated the percentage of solid particles remaining on the steel surface after contacting the silicon surface by using a pressure-controlled touch machine. Comparing their results with ours, they have a decreasing trend as the number of contacts increases. The transfer process in Zhao’s study is mainly in the first three steps, while, in our study, the process is mainly in the first five steps. We assumed this was because Zhao's experiment was conducted using solid particles as the simulant, while our experiment was conducted using fluorescent solution. This discussion has been added in the 3rd paragraph of Section “3.3. Analysis of the variation law of the ground fluorescence value with walking”.

References:

[1] Zhao, P.; Li, Y.; Tsang, T.; Chan, P. Equilibrium of particle distribution on surfaces due to touch. Building and Environment, 2018, 143, 461–472.

Reviewer 2 Report

The authors describe the use of a fluorescent solution to replicate pathogen droplets on the ground, as well as optical analysis to conduct propagation and diffusion studies for liquid pollutants. These tests are predicated on pathogen transmission from shoe soles to indoor ground through. The effects of body weight, walking frequency, and solution viscosity, are thoroughly investigated. The diffusion law is established, and the elements that influence this sort of diffusion mode are clarified. The current work is intriguing, and I urge that it be published after minor revision. Here are my thoughts:

  • The photos in Figure 4 are hazy.
  • The y-axis caption of plot 4 should be edited.
  • The equality sign is missing in the first section of equation 3.
  • The fitting equations should be supplied on the plots in figure 7 and delete table 3.
  • The study's purpose should be stressed and stated explicitly.
  • Other analytic experimental procedures should be used to explain and emphasize the statistically generated findings.

Author Response

We would like to thank all the reviewers for the careful review and the valuable comments, which allowed us to improve the paper. All changes in the revised paper were highlighted by yellow background. Please see the attachment.

Reviewer 3 Report

Referee

Title: Propagation and diffusion of fluorescent substances with foot-prints in indoor environments  

Authors: Manman Ma, Hao Han, Ziwang Zhao, Yuxiao Sun, Yuanqi Jing, Lei Wang  and Fei Li

Line 203

FIG.4. IT IS NOT CLEAR ENOUGH WHAT REPRESENTS IMAGES 5-11

Please clarify

Fig. 6 line 249:

PLEASE INSERT THE MEASUREMENT UNITS FOR THE AXIS. THE LEGEND OF THE FIGURES IS NOT ENOUGH LISIBLE

Conclusions

THE CONCLUSIONS ARE CONSISTENT.

 SHOULD MORE UNDERLINE THE IDEA THAT THE RESEARCH SHOULD BE STILL DEVELOPED AND TESTED AND WITH MORE SUBJECTS IN VARIOUS CLIMATIC ENVIRONMENTS, FURTHER TO OPTIMIZE AN ENVIRONMENTALLY ADAPTIVE DETECTION ALGORITHM-PROVIDES A NEW METHOD FOR EXAMINING THE CONTACT PROPAGATION AND DIFFUSION OF GROUND POLLUTANTS WHILE WALKING.

THE REFERENCES SHOULD BE COMPLETED (E.G. DOI IS MISSING)

PLEASE ENLARGE AND INTERNATIONALIZE AND UPDATE TO 2022 THE REFERENCES LIST.

WE PROPOSE YOU, AS POTENTIAL RELEVANT FOR THE PAPER, TO SEE AND CITE ALSO:

ILIEȘ, D.C.; MARCU, F.; CACIORA, T.; INDRIE, L.; ILIEȘ, A.; ALBU, A.; COSTEA, M.; BURTĂ, L.; BAIAS, Ș.; ILIEȘ, M.; SANDOR, M.; HERMAN, G.V.; HODOR, N.; ILIEȘ, G.; BERDENOV, Z.; HUNIADI, A.; WENDT, J.A. INVESTIGATIONS OF MUSEUM INDOOR MICROCLIMATE AND AIR QUALITY. CASE STUDY FROM ROMANIA. ATMOSPHERE 202112, 286. HTTPS://DOI.ORG/10.3390/ATMOS12020286

MARCU, F.; HODOR, N.; INDRIE, L.; DEJEU, P.; ILIEȘ, M.; ALBU, A.; SANDOR, M.; SICORA, C.; COSTEA, M.; ILIEȘ, D.C.; CACIORA, T.; HUNIADI, A.; CHIȘ, I.; BARBU-TUDORAN, L.; SZABO-ALEXI, P.; GRAMA, V.; SAFAROV, B. MICROBIOLOGICAL, HEALTH AND COMFORT ASPECTS OF INDOOR AIR QUALITY IN A ROMANIAN HISTORICAL WOODEN CHURCH. INT. J. ENVIRON. RES. PUBLIC HEALTH 202118, 9908. HTTPS://DOI.ORG/10.3390/IJERPH18189908

What is the main question addressed by the research?

The main question addressed is of high importance for public healthThe paper try to optimize an environmentally adaptive detection algorithms concerning the Propagation and diffusion of fluorescent substances with foot prints in indoor environments. The paper review of relevant range of articles in this research domain etc

Is it relevant and
interesting?

The paper is relevant for public health in postpandemic period in complex indoor environments. It synthetizes the actual available literature data, focus mainly on efficiency of devices based on tests conducted in laboratory conditions. It helps to prevent wasted time and money by optimizing variables caused by false alarms regarding the contaminants pollution diffusion.

How original is the topic?

Is an actual topic with high degree of originality; it is important subject especially in the post pandemic period, authors findings confirmed that false alarms did not occur in situations where no alarm was unnecessary.

What does it add to the subject
area compared with other published material?

The paper is well documented because the authors cited more than 22 scientific published articles, updated to 2022. And because the literature in this specific experimental field is not very well represented, the paper can be considered innovative level.

Is the paper well written?

The paper is well written. In my opinion the quality of English translation is good.

Is the text clear and easy to read?

The text is well structured, clear and easy to read from the specialists in the field but as well as from the persons from public.

Are the conclusions consistent with the evidence and arguments presented?

The conclusions are consistent but should more underline the idea that the research should be still developed and tested in various climatic environments and with more subjects, further to optimize an environmentally adaptive detection algorithm-provides a new method for examining the contact propagation and diffusion of ground pollutants while walking. 

6 May 2022

Author Response

(The authors gave the same response as above.)

Reviewer 4 Report

Comments on

Propagation and diffusion of fluorescent substances with footprints in indoor environments

Manuscript Number: IJERPH-1733747

General comments

The study used fluorescence technology to study the specific mode of transmission of pathogens. Viruses carried in the soles of people's shoes can be carried elsewhere by walking, and the main influencing factors were analyzed. This study has certain reference value for the prevention and control of the new coronavirus. However, there are still some shortcomings in this study, which need to be revised. There are the following main problems:

a: The number of volunteers is too small, only three; whether the center of gravity of the left and right feet of the participants in the experiment is considered to be different;

b: The influencing factors considered are all related to people, but the number and type of pathogens in the sole may also have a greater impact on the transfer efficiency, etc.;

c: The conclusions are too complex and require significant revision, highlighting how innovative your research is, what scientific problem it addresses, and the main argument.

Specific comments

Here are some detailed comments listed as follows, but not complete:

Pictures of fluorescence experiment results are best in color;

More tables reflecting your experimental results should be included in the supporting materials;

Too many keywords; need to highlight the unique mode of transmission of the pathogen you are studying;

Line 213-219: Normal people have two feet. It is not recommended that you analyze one person's two feet as two independent samples;

Line 231-240: If the fluorescence intensity, that is, the number and concentration of pathogens, reaches a lower level after five or six steps, what is the significance of your research, please explain;

Line 329-339: What is the transmission and diffusion law of liquid pollutants during walking, please explain.

Author Response

There are the following main problems

(a): The number of volunteers is too small, only three; whether the center of gravity of the left and right feet of the participants in the experiment is considered to be different;

Reply

Thank you for your comments. In this study, the three volunteers were employed to represent three weight intervals (50-55 kg, 60-65 kg and 70-75 kg, according to the Report on Nutrition and Chronic Diseases in China). This does not mean that we have only three cases or trials. Nine the experimental cases were designed by the orthogonal test, and for each case, six replicate experiments were conducted to ensure the reliability of the data. In total, 54 sets of data were used for analysis. This explanation is placed in the first and third paragraphs of Section “2.4. Experimental case design”.

In terms of the effect of the gravity center, this study used a paired samples t-test to determine whether there was a statistically significant difference in the amount of fluorescent solution on the floor between the three volunteers' left and right foot walking differences. The results showed that none of the results showed a significant effect at the 0.05 significance level. Therefore, we assumed the change in center of gravity was not an influencing factor. This explanation is placed in the second paragraph of Section “3.2. Analysis of the independence of each foot”.

(b): The influencing factors considered are all related to people, but the number and type of pathogens in the sole may also have a greater impact on the transfer efficiency, etc.;

Reply:

Thank you for your comments. As you said, the type of pathogen, its survival time, etc. may have effects on the transmission efficiency. However, due to the limitation of the experimental conditions, it is difficult to use the pathogens to conduct the experiment and analyze effects of their number and type. Therefore, this study only considered the influence of external factors such as body weight, walking frequency, and solution viscosity by using fluorescent substances and investigated its propagation and diffusion law with footprints. This discussion has been added in the last paragraph of Section “4. Conclusions”.

(c): The conclusions are too complex and require significant revision, highlighting how innovative your research is, what scientific problem it addresses, and the main argument.

Reply:Thank you very much for your suggestion, we have revised Section “4. Conclusions” as follows:

“This study used a fluorescent solution to simulate the pollutant propagation during walking in an indoor environment and explored the propagation and diffusion law of ground pollutants with footprints. The effects of body weight, walking frequency, and solution viscosity on the footprint decay (D) and transfer efficiency (τ) of the fluorescent solution were studied via the fluorescent footprint imaging. The conclusions are follows:

(1). As the number of walking steps increased, the amount of fluorescent substance transferred onto the experimental floor decreased exponentially, the fluorescence value begins to fluctuate after the fifth step and decreases to ~6% of the initial value after the fifth step.

(2). The relationship between the normalized gray value of the fluorescent solution (D) on each floor panel i was , it can be used to estimate the number of pollutants on the ground…

(3). The transfer efficiency was distributed within a range of 0.51–0.72 for all experiments, and only the body weight caused it to change significantly…”.

Specific comments

Here are some detailed comments listed as follows, but not complete:

(1)、Pictures of fluorescence experiment results are best in color;

Reply

The image is processed in grayscale to calculate its fluorescence intensity better, and the fluorescence image has been also made clearer by increasing the contrast.

(2)、More tables reflecting your experimental results should be included in the supporting materials;

Reply: Thank you for your suggestion. The averaged normalized gray values of the fluorescent solution with the number of walking steps corresponding to Figure 6 has been added in the supporting information.

(3)、Too many keywords; need to highlight the unique mode of transmission of the pathogen you are studying;

Reply

Thank you very much for your suggestion. The keywords have been revised as: “Fluorescent substances; Pathogens; Ground pollutant diffusion; Footprint”.

The unique mode of transmission of the pathogen has been highlighted in Section “Abstract”: “Some studies have shown that contaminants can be transferred between floors and the soles, and there are few studies on pollutant propagation and diffusion caused by human walking in real-life situations. This study explored the propagation and diffusion law of ground pollutants with footprints during indoor walking through employing a fluorescent solution as a simulant.”.

(4)、Line 213-219: Normal people have two feet. It is not recommended that you analyze one person's two feet as two independent samples;

Reply:

Thank you very much for your suggestion. The statement of “independent samples” is not appropriate. In our study, the paired sample t-test was employed to determine whether there was a statistically significant difference in the amount of fluorescent solution on the floor panels between the data collection conditions and the left- and right-foot walking characteristics of the participants. It was found that the difference between the left and right feet had no statistically significant effect on the amount of fluorescent solution transferred to the floor panel (P > 0.05); therefore, the walking difference between the left and right foot was not considered as an influencing factor during the data analysis. This revision has been added in the second paragraph of Section “3.2. Analysis of the independence of each foot”.

(5)、Line 231-240: If the fluorescence intensity, that is, the number and concentration of pathogens, reaches a lower level after five or six steps, what is the significance of your research, please explain;

Reply

Thank you for your comment. Until this study was conducted, the law of pathogen transmission and diffusion with the footprint could not be clarified. Based on this study, we determined the propagation and diffusion law of liquid pollutants during walking and found that the fluorescence value begins to fluctuate after the fifth step and decreases to ~6% of the initial value after the fifth step. If we then knew the human step data, the footprint transmission distance of contaminants could be deduced. And these conclusions have guiding significance for pollutant prevention and control. This is also the purpose and significance of this study. This discussion has been added in the last paragraph of Section “4. Conclusions”.

(6)、Line 329-339: What is the transmission and diffusion law of liquid pollutants during walking, please explain.

Reply

This study found that the amount of fluorescent substances on the ground tended to decay exponentially as walking steps increased, and the footprint contaminant propagation is mainly through the first five steps. The relationship between the footprint contaminant and the number of walking steps was established. It was also found that the transfer efficiency was mainly in the range of 0.51-0.72, and only body weight had a significant effect on both the transfer efficiency. This explanation is in Section “Abstract” and “4. Conclusions”.

Reviewer 5 Report

It is not clear how and why this research is necessary. The reader must be contextualized about the problem and the main contributions of the work must be clearly presented.

The results must be compared with similar work in the state of the art and a depth discussion must be made to clearly identify the novel findings in this work. In a present manner, this work can not support future research in this field. Moreover, the authors should provide a detailed setup of the experiments in order to enable the readers to reproduce the experiments.

Most important, it is not clear how and why a case study can be validated using 3 volunteers. This seems a critical point in this study. Furthermore, if not supported and justified by scientific evidence I think this is a critical limitation.

Author Response

Response to Reviewer #5’s comments

(1)、It is not clear how and why this research is necessary. The reader must be contextualized about the problem and the main contributions of the work must be clearly presented.

Reply:

Thank you for your comments. The study's necessity and contribution has been stressed and stated explicitly in Section “Abstract”: “Some studies have shown that contaminants can be transferred between floors and the soles, and there are few studies on pollutant propagation and diffusion caused by human walking in real-life situations. This study explored the propagation and diffusion law of ground pollutants with footprints during indoor walking through employing a fluorescent solution as a simulant. The footprint decay (D) and transfer efficiency (τ) of the fluorescent solution transferred from the sole to the indoor floor during walking were analyzed based on the fluorescent footprint imaging. And the effects of namely body weight, walking frequency, and solution viscosity were also investigated.”.

It has also been stressed and stated in Section “1. Introduction”: “…there is a possibility that the detected pathogens were brought into the pharmacy on the shoes of medical staff, ultimately propagating their spread. However, to date, this pathogen propagation mode has not received particular attention…In this study, the propagation and diffusion law of ground pollutants with footprints during indoor walking were investigated through employing a fluorescent solution and footprint imaging…The influence of three factors, namely body weight, walking frequency, and solution viscosity on the footprint decay and transfer efficiency were examined in detail. Finally, the diffusion law was determined and the factors influencing this type of diffusion mode are clarified.”

(2)、The results must be compared with similar work in the state of the art and a depth discussion must be made to clearly identify the novel findings in this work. In a present manner, this work cannot support future research in this field. Moreover, the authors should provide a detailed setup of the experiments in order to enable the readers to reproduce the experiments.

Reply

Thank you for your comments. Most of the studies on contact propagation have focused on hand to various surfaces. For the similar work, Zhao et al. [1] investigated the percentage of solid particles remaining on the steel surface after contacting the silicon surface by using a pressure-controlled touch machine. Comparing their results with ours, they have a decreasing trend as the number of contacts increases. The transfer process in Zhao’s study is mainly in the first three steps, while, in our study, the process is mainly in the first five steps. We assumed this was because Zhao's experiment was conducted using solid particles as the simulant, while our experiment was conducted using fluorescent solution. There are few studies on the propagation law of footprints. Although some studies have investigated pollutant propagation between shoe soles and the ground, the studies on pollutant propagation and diffusion caused by human walking in real-life situations is lacked. This discussion has been added in the 3rd paragraph of Section “3.3. Analysis of the variation law of the ground fluorescence value with walking” and the 2nd paragraph of Section “1. Introduction”.

For the detailed setup of the experiments: 1) Selection and preparation of the fluorescent solution (Section 2.1); 2) Selection of the experimental floor and shoes (Section 2.2); 3) Fluorescence data collection methods (Section 2.3); 4) Experimental case design (Section 2.4). We have reorganized the experimental steps, and we think the readers can reproduce the experiments. In addition, we also have added the experimental data in the supporting information for the reader's reference.

References:

[1] Zhao, P.; Li, Y.; Tsang, T.; Chan, P. Equilibrium of particle distribution on surfaces due to touch. Building and Environment, 2018, 143, 461–472.

(3)、Most important, it is not clear how and why a case study can be validated using 3 volunteers. This seems a critical point in this study. Furthermore, if not supported and justified by scientific evidence I think this is a critical limitation.

Reply

Thank you for your comments. In this study, the three volunteers were employed to represent three weight intervals (50-55 kg, 60-65 kg and 70-75 kg, according to the Report on Nutrition and Chronic Diseases in China). This does not mean that we have only three cases or trials. Nine the experimental cases were designed by the orthogonal test, and for each case, six replicate experiments were conducted to ensure the reliability of the data. In total, 54 sets of data were used for analysis. This explanation is placed in the first and third paragraphs of Section “2.4. Experimental case design”.

Round 2

Reviewer 4 Report

Your work is interesting. Hope you keep trying.

Author Response

Thank you for your encouragement.

Reviewer 5 Report

I am not satisfied with the authors' revision to my previous comments:

(X1)、The results must be compared with similar work in the state of the art and a depth discussion must be made to clearly identify the novel findings in this work. In the present manner, this work cannot support future research in this field. 

The comparison must be done with several studies not only with one.

(X2) Most important, it is not clear how and why a case study can be validated using 3 volunteers. 

If there is no scientific evidence of validation methods using only 3 volunteers I do not think that this experiment can be accepted. The experimental setup is very limited.
